# Cell Morphology-Guided Small Molecule Generation with GFlowNets

**Stephen Zhewen Lu** [1 2]  **Ziqing Lu** [3]  **Ehsan Hajiramezanali** [3]  **Tommaso Biancalani** [3]  **Yoshua Bengio** [1 4]
**Gabriele Scalia** [3]  **Michał Koziarski** [1 4]

## Abstract

High-content phenotypic screening, including high-content imaging (HCI), has gained popularity in the last few years for its ability to characterize novel therapeutics without prior knowledge of the protein target. This work focuses on the novel task of HCI-guided molecular design. We consider an approach in which we leverage an unsupervised multimodal joint embedding to define a latent similarity as a reward for GFlowNets. The proposed model learns to generate new molecules that could produce phenotypic effects similar to those of the given image target, without relying on pre-annotated phenotypic labels. We demonstrate that our method generates molecules with high morphological and structural similarity to the target, increasing the likelihood of similar biological activity.

## 1. Introduction

Phenotypic screening allows assessing drug efficacy based on observed biological effects, without detailed knowledge of the underlying mode of action. High-content phenotypic screening, especially through high-content imaging (HCI) (Bray et al., 2016), provides rich data such as morphological changes in cell shape and structure. This can potentially elucidate broad biological effects, targets, and mechanisms of action (Moffat et al., 2017). Despite its potential to guide the drug discovery process, challenges persist in effectively utilizing this data.

One common approach to utilize this information is training a supervised classification model to recognize the phenotype of interest, and then use it to virtually screen existing libraries (Krentzel et al., 2023). However, this can require a substantial amount of labeled data and prior knowledge

of the phenotype of interest, often unavailable for poorly understood diseases. A preferred scenario would be to have a method capable of designing new compounds based on only a few, or even a single, target morphological readout. Another challenge lies in the screening process itself. Virtual screening limits the search space to existing screening libraries, which are significantly smaller than the whole drug-like space, often estimated to contain more than $10^{60}$ (Lipinski et al., 1997) molecules. This limitation can become particularly important when structurally and functionally novel molecules are desirable, for example, to improve the potency and diversity of the leads or to overcome unwanted secondary effects (Jain et al., 2022; Gao et al., 2024; Song & Li, 2023; Ghari et al., 2023).

In this paper, we address both of these challenges by proposing a generative method guided by the target cell morphology. Specifically, we propose a reward function that utilizes the latent similarity between the generated molecule and the target morphology. For this, we utilize a multi-modal contrastive learning model that aligns small molecules to morphological readouts. Then, we use this reward to train a Generative Flow Network (GFlowNet), as recently proposed by Bengio et al. (2023), to generate molecules capable of inducing similar morphological outcomes. We demonstrate that our approach is capable of generating diverse molecules with high latent similarity to the provided morphology, which translates into a higher likelihood of obtaining similar predicted biological activity. We also show that the generation process can be structurally conditioned by using joint latent embeddings, which combine both target readout and molecule, further improving the performance. Practical use cases of the proposed method include generating molecules that induce a desired cell morphology obtained through gene perturbations (Rohban et al., 2022), scaffold hopping, where novel molecular structures with similar effects to a target one are desired (Hu et al., 2017), and, more generally, molecular design guided by phenotypic readouts (Krentzel et al., 2023).

## 2. Related work

**Generative models for drug discovery.** There is a plethora of methods for molecular generation in the literature (Mey-

---

[1]Mila – Québec AI Institute [2]McGill University [3]Biology Research | AI Development (BRAID), Genentech [4]Université de Montréal. Correspondence to: Michał Koziarski <michal.koziarski@mila.quebec>.

*Accepted at the 1st Machine Learning for Life and Material Sciences Workshop at ICML 2024.* Copyright 2024 by the author(s).

ers et al., 2021; Bilodeau et al., 2022). They can be broadly categorized based on the molecular representation used: textual representation such as SMILES (Kang & Cho, 2018; Arús-Pous et al., 2020; Kotsias et al., 2020), molecular graphs (Jin et al., 2018; Maziarka et al., 2020; Pedawi et al., 2022; Diamant et al., 2023) or 3D atom coordinate representations (O Pinheiro et al., 2024); as well as the underlying methodology, e.g. variational autoencoders (Jin et al., 2018; Maziarka et al., 2020), reinforcement learning (RL) (Loeffler et al., 2024) or diffusion models (Runcie & Mey, 2023; Uehara et al., 2024). These methods have found applications in drug discovery, reporting successes in several areas such as immunology and infectious diseases (Godinez et al., 2022; Moret et al., 2023). Recently, Generative Flow Networks (GFlowNets) (Bengio et al., 2021; Nica et al., 2022; Shen et al., 2023; Volokhova et al., 2024; Koziarski et al., 2024a; Gaiński et al., 2024; Koziarski et al., 2024b) emerged as a promising paradigm for molecular generation due to the ability to sample diverse candidate molecules, which is crucial in the drug discovery process. Importantly, similar to RL, GFlowNets can be trained based on the specified reward function, which makes them suitable for phenotypic discovery. Compared to existing methods, which focus on conditional generation based on a single property or multiple properties of interest, we tackle generation guided by high-content readouts, which we achieve through a multimodal latent joint representation.

**Deep learning for high-content molecular perturbations.**

High-content phenotypic screening, particularly high-content imaging (HCI), has become crucial in drug discovery for characterizing molecular effects in cells and elucidating targets, gene programs, and biological functions (Moffat et al., 2017; Chandrasekaran et al., 2021). Recent advances propose deep learning techniques to accelerate and enhance these processes (Gavriilidis et al., 2024). Predictive models to infer the outcome of molecular effects have been developed both for transcriptomic readouts (Lotfollahi et al., 2019; Hetzel et al., 2022; Piran et al., 2024) and HCI readouts (Palma et al., 2023). In these models, the output is highly multi-dimensional, capturing the full readout of the high-throughput experiment, thus potentially requiring a large amount of data for training and making it challenging to separate biological effects from background signals. In contrast to these works, we focus on the inverse problem, designing molecules leading to a specific (target) readout.

**Joint representation of molecules and high-content readouts.** Multi-modal contrastive models (Radford et al., 2021) have been used to align molecular representations to high-dimensional readouts in latent space, thus capturing shared features while avoiding high-dimensional supervised losses (Sanchez-Fernandez et al., 2022; Zheng et al., 2022; Nguyen et al., 2023). While these models can be used for screening

tasks, reporting improved generalization ability compared to molecule-only models, they are unable to perform generative tasks. Compared to existing works in this area, we focus on the novel task of HCI-guided molecular design, while relying on a joint representation to guide the generation.

## 3. Method

### 3.1. Generative Flow Networks

GFlowNets are amortized variational inference algorithms that are trained to sample from an unnormalized target distribution over compositional objects. GFlowNets aim to sample objects from a set of terminal states $\mathcal{X}$ proportionally to a reward function $\mathcal{R} : X \to \mathbb{R}^+$. GFlowNets are defined on a pointed directed acyclic graph (*DAG*), $G = (S, A)$, where:

- $s \in S$ are the nodes, referred to as states in our setting, with the special starting state $s_0$ being the only state with no incoming edges, and the terminal states $\mathcal{X}$ have no outgoing edges,

- $a = s \to s' \in A$ are the edges, referred to as actions in our setting, and correspond to applying an action while in a state $s$ and landing in state $s'$.

A state sequence $\tau = (s_0 \to s_1 \to \cdots \to s_n = x)$, with $s_n = x \in \mathcal{X}$ and $a_i = (s_i \to s_{i+1}) \in A$ for all $i$, is called a complete trajectory. We denote the set of trajectories as $\mathcal{T}$.

**Trajectory balance.** Several training losses were explored to train a GFlowNet. Among those, trajectory balance (Malkin et al., 2022a) has been shown to improve credit assignment. In addition to learning a forward policy $P_F$, we also learn a backwards policy $P_B$ and a scalar $Z_\theta$, such that, for every trajectory $\tau = (s_0 \to s_1 \to \cdots \to s_n = x)$, they satisfy:

$$Z_\theta \prod_{t=1}^{n} P_F(s_t|s_{t-1}) = R(x) \prod_{t=1}^{n} P_B(s_{t-1}|s_t) \quad (1)$$

### 3.2. Multi-modal contrastive learning

Contrastive learning is a self-supervised approach that learns embeddings by maximizing agreement between similar samples and minimizing it between dissimilar ones, using contrastive loss functions like InfoNCE (Oord et al., 2018). Multi-modal contrastive learning effectively learns joint representations from diverse data modalities. A notable instance of this approach is CLIP (Radford et al., 2021), which aligns textual descriptions with visual representations, enabling robust cross-modal understanding. We leverage a multi-modal contrastive model to learn a joint embedding

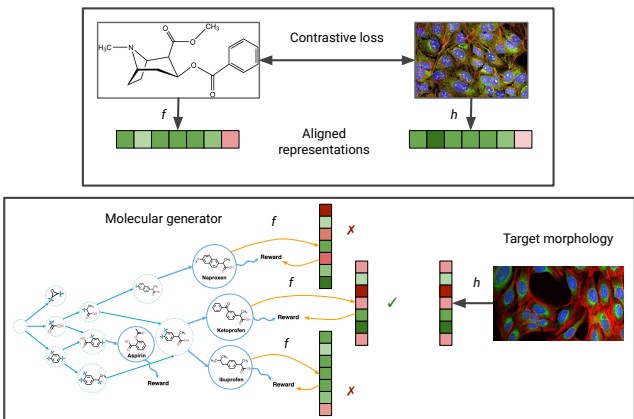

*Figure 1.* Graphical illustration of the proposed approach. In the first stage (top), cross-modal contrastive learning is used to train latent encoders $f$ and $h$ that produce aligned representations between molecules and readouts. Then, in the second stage (bottom), the target morphology readout is first converted into a latent vector, and the similarity between that and latents from molecules output by a generator is used as a reward. The model learns to sample molecules capable of inducing similar latents to the target.

of molecules and molecular effects. This choice avoids high-dimensional supervised losses and promotes learning "informative" features for the task (i.e., features that relate the two modalities to each other).

Let $\{(x_i, y_i)\}_{i=1}^{N}$ be a batch of $N$ pairs of molecular graphs $(x_i)$ and their corresponding morphology images $(y_i)$. Let $f$ and $h$ be the molecular and morphology encoders, respectively. Let the similarity between the molecular graph and morphology image embeddings be defined as $S_{ij} = \exp\left(\frac{\text{sim}(f(x_i), h(y_j))}{\tau}\right)$, where $\text{sim}(f(x_i), h(y_j))$ denotes the cosine similarity between embeddings. The CLIP loss is defined as follows:

$$L_{\text{CLIP}} = \frac{1}{N} \sum_{i=1}^{N} \left[ -\log \frac{S_{ii}}{\sum_{j=1}^{N} S_{ij}} - \log \frac{S_{ii}}{\sum_{j=1}^{N} S_{ji}} \right], \tag{2}$$

where $\tau$ is a temperature parameter.

Instead of relying on CLIP, in this work, we leverage the closely related Geometric Multimodal Contrastive (GMC) loss (Poklukar et al., 2022). GMC leverages the same loss function in (2) to align modality-specific encoders to a joint encoder, which takes as input all modalities. Therefore, in addition to providing modality-specific embeddings $f$ and $h$, it also provides a *joint* embedding $fh$ that we leverage when both the target molecule and its readout are available.

### 3.3. GFlowNets for morphology-guided molecular design

The proposed approach exploits recent developments in multi-modal contrastive learning and molecular generation with GFlowNets into a unified pipeline, as illustrated in Figure 1. The method relies on first training a contrastive learning model capable of producing aligned latent repre-

sentations, and then using these representations as a guiding signal for GFlowNet. We define the reward function for the GFlowNet from the embeddings of the GMC model as described in Section 3.2:

$$R(x|y) = 1 + \frac{f(x)h(y)}{2\|f(x)\|\|h(y)\|}. \tag{3}$$

This normalized cosine similarity between the target morphology latent and the generated structure latent is crucial to enforce the non-negativity of the GFlowNet reward. During the training of the GMC model, we impose early-stopping using the correlation between the cross-modality distance metric. We observe that early-stopping reduces the variance of cosine similarity between the multimodal GMC embeddings, while it does not affect the performance of GFlowNet (see Appendix A.4 for more details). Inspired by using replay buffer in reinforcement learning (Vemgal et al., 2023), we leverage replay buffer with known decomposed structure when training on joint morphology and structure-guided generations. This increases the structural similarity of generated samples to the known target.

## 4. Experimental study

We evaluate the proposed approach, first by examining the quality and diversity of generated samples, and then their predicted biological activity in downstream tasks. Experiments verifying the underlying assumption of our method, namely that the similarity in latent space produced by the contrastive model correlates with the morphological similarity, can be found in Appendix A.1. Experiments on structurally conditioned generation, where we base our target latent on the combination of target morphology and the associated molecular structure, can be found in Appendix A.2.

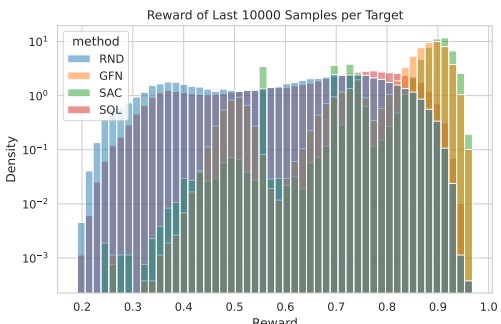 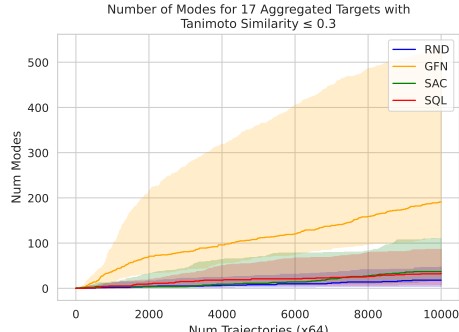

*Figure 2.* Comparison of examined methods in terms of reward optimization: a) distribution of rewards from generated samples, 10,000 samples generated by a trained model (left) and b) number of modes discovered (right). Both plots aggregate across all examined targets.

## 4.1. Set-up

We perform experiments on the Cell Painting dataset introduced by (Bray et al., 2016; 2017), as further pre-processed in (Moshkov et al., 2023). The dataset includes 16,170 molecules and associated cell morphology images. Each image includes five color channels that describe the morphology of five cellular compartments. Images have been pre-processed using CellProfiler 3.0 (McQuin et al., 2018; Moshkov et al., 2023).

Additionally, to support the evaluation of the generated molecules, we leverage oracle models independently trained on data from multiple assays released by the Broad Institute (Moshkov et al., 2023) (see Appendix B.3 for more details). This allows us to evaluate the ability of the model to generate molecules with biochemical and cellular effects similar to the (unknown) target molecule.

## 4.2. Generating high reward and diverse samples

We are interested in evaluating the capabilities of a GFlowNet in optimizing the specified reward function. Given our primary focus on its application in the initial discovery stage, it is essential to generate not only high-reward outcomes but also a diverse set of samples.

We compare GFlowNet with random sampling (RND) and two standard RL-based baselines: soft Q-learning (SQL) (Haarnoja et al., 2017) and soft actor-critic (SAC) (Haarnoja et al., 2018). Note that since, to the best of our knowledge, this is the first published attempt at guiding the generative molecular model with expected image morphology outcome, we focus specifically on benchmarking against other potential molecular generation methods.

We show the distribution of rewards for generated samples and the number of discovered modes (defined as molecules with reward ≥ 90th percentile and Tanimoto similarity to other modes ≤ 0.3) in Figure 2, and the distribution of similarities between the top-100 generated samples in Figure 3 (each figure aggregated across all considered targets).

As shown, GFlowNet learns to sample high-reward candidate molecules (with a significantly higher average reward than random sampling and SQL, comparable to SAC) while also significantly improving the diversity compared to SAC (with a lower similarity between top candidates). Both of the above translate into a significantly higher number of discovered modes than the baseline methods. The diversity of the generated samples is further illustrated in Figure 3, where a UMAP visualization of the molecular structures produced by different methods is presented for a specific target. As can be seen, GFlowNet displays sample coverage similar to random sampling, which is a desirable outcome.

## 4.3. Biological activity estimation

So far, we have established that the proposed approach is capable of generating diverse samples with high reward and that there is a moderate correlation between the reward and the similarity to the target. However, what is critical in the end is whether this will translate into generated molecules inducing similar biological effects to the original target perturbation (ground truth). Ideally, we wish to evaluate this on the basis of experiments comparing generated molecules and the ground truth, but this can be costly and time-consuming. Instead, in the following we estimate the similarity of the biological effect based on the available data. We consider two approaches.

First of all, we would expect a perfectly optimized generator to be able to sample known molecules that induced target morphological profiles. In practice, this might be unattainable: not only is the problem itself heavily under-constrained (we expect multiple molecules, likely a very large number, to be able to induce a given morphological outcome), but also our morphological similarity estimation is intrinsically noisy. Because of the above, what we try to achieve is the highest possible structural similarity of generated samples to the known molecule that induced the given morphology. The maximum Tanimoto similarity to the known molecular target, averaged across all considered targets, is presented in Table 1. As shown, the proposed

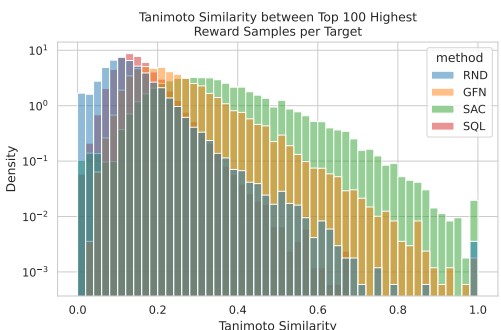
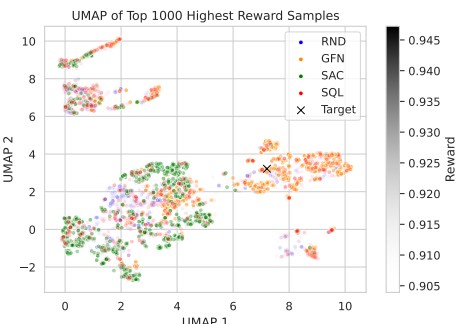

*Figure 3.* Comparison of examined methods in terms of diversity: a) distribution of Tanimoto similarities between top-100 generated samples (left; aggregated across all examined targets, lower is better) and b) structural diversity of top-1000 highest reward samples to Target #8636 (right).

*Table 1.* Max. Tanimoto similarity to the target in the last 10,000 samples, averaged across all targets.

|  | Random | Soft Actor-Critic | Soft Q-Learning | GFlowNet |
|---|---|---|---|---|
| Morphology Target | 0.305 (± 0.057) | 0.261 (± 0.068) | 0.329 (± 0.079) | **0.337 (± 0.092)** |
| Joint Target | 0.311 (± 0.064) | 0.388 (± 0.163) | 0.309 (± 0.064) | **0.451 (± 0.163)** |

approach generally recovers the underlying targets more effectively, suggesting its utility in identifying molecules with expected biological activity. Unsurprisingly, this effect is particularly pronounced when conditioning on specific molecular structures.

The second approach we consider involves utilizing oracle models for predicting biological activity. We train an MLP using molecular structures, specifically their extracted molecular fingerprints, as inputs to predict the outcomes of biological assays (details of the model training are provided in Appendix B.4). For each target molecule, at least one associated assay has a positive outcome. The objective is to generate molecules with a high predicted probability of producing a positive outcome in the specific assay for which the target molecule has known activity. The number of generated samples with high predicted assay probability ($\geq 0.7$) is presented in Figure 4. Note that, due to the high uncertainty of the oracle model, we are primarily interested in quantifying the number of high-likelihood samples. As can be seen, using guided generation helps improve the proportion of generated molecules with high predicted assay probability, which serves as a proxy for generating molecules with similar biological activity. It is worth noting that while we observe a higher predicted probability for SAC when considering the top-1000 molecules by reward, this trend reverses when considering different modes, once again highlighting the higher diversity combined with high reward of GFlowNet samples. It is also worth noting that we observe a large variance across targets. While this can be partially attributed to the uncertainty of the oracle, further investigation of factors determining whether the proposed approach is helpful or not for a given target is an important

future research direction.

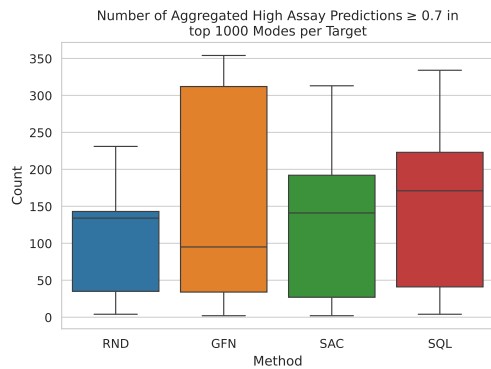

*Figure 4.* Number of samples with predicted assay probability $\geq$ 0.7 in the top 1000 modes with Tanimoto similarity $\leq 0.3$. Results are aggregated across multiple targets.

## 5. Conclusions

In this paper, we consider the task of designing a generative model able to produce molecules that can induce a cell morphology profile similar to a given target. Such framework is broadly applicable, for example to design drugs mimicking the effect of a genetic perturbation, designing drug analogs, or, more generally, molecular design guided by phenotypic readouts. The proposed approach relies on the GFlowNet framework for molecular generation, and uses a reward based on the latent similarity of representations from a multi-modal contrastive learning model. To the best of our knowledge, this is the first published attempt at the challenging task of guiding the generative molecular model with the expected image morphology outcome. We experimentally

demonstrate the usefulness of the proposed approach for generating diverse drug candidates, which was shown to increase the likelihood of producing molecules with similar biological activity when compared to random screening.

## Acknowledgements

The research was supported by funding from CQDM Fonds d'Accélération des Collaborations en Santé (FACS) / Acuité Québec and Genentech.

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

# A. Additional experiments and plots

## A.1. Relation between proposed reward and morphological distance

The underlying assumption behind our method is that the latent representations produced by the contrastive learning model are of a high enough quality to reliably compute similarity. Specifically, given a target pair of a molecule and its associated morphology $(\dot{x}, \dot{y})$, an arbitrary pair of a generated molecule and the morphology that would be induced by that molecule $(x, y)$, distance $d$ in the input modality space, distance $\hat{d}$ in the latent space, and models $f$ and $h$ that transforms our input into a latent representation, we require that

$$R(x|\dot{y}) = \hat{d}(h(\dot{y}), f(x)) \sim d(\dot{y}, y). \qquad (4)$$

It is worth noting that during generation we always know $x$, which is simply the molecule generated by our model, and never know $y$, which is why we need $f$. Similarly, we always know the target morphology $\dot{y}$, and in some settings, such as finding drug analogs, we might also have access to the associated molecule $\dot{x}$. In particular, in the latter setting, we might consider conditioning on the joint latent, produced based on the pair of $(\dot{x}, \dot{y})$: $R(x|\dot{x}, \dot{y}) = \hat{d}(fh(\dot{x}, \dot{y}), f(x))$.

To evaluate this assumption, for every pair of observations from the dataset, we measure the correlation between the similarity of morphological features and the similarity between the latent representation of the structure from the first observation and the latent representation of the target morphology from the second observation. Additionally, we also do the same for the joint target latent, computed based on both the morphology and the associated molecule structure. The results are presented in Figure 5. As can be seen, in both cases we observe a medium level of correlation. While not perfect, we argue that this can be sufficient for screening purposes (where we are interested in improving the hit ratio, but due to the difficulty of the task do not expect very precise outcomes).

## A.2. Joint morphology and structure-guided generation

Our original assumption is that generating a molecule capable of inducing specified morphological outcome would be sufficient to produce desirable drug candidates. However, it is worth noting that in practice this problem can be significantly under-constrained, meaning that there might be a large number of not sufficiently druglike, too toxic, or otherwise not useful molecules capable of inducing the

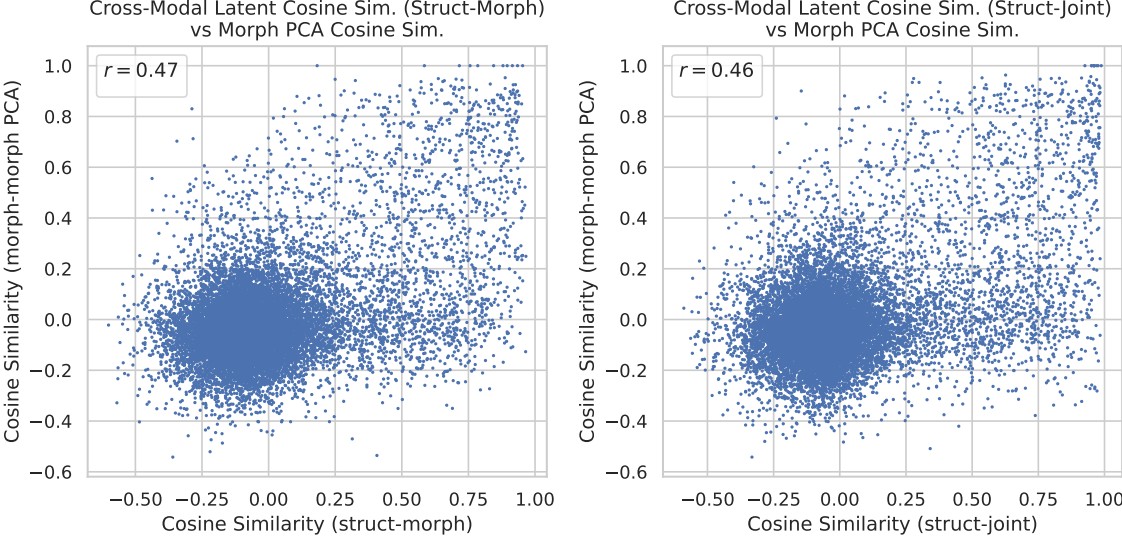

Figure 5. Correlation between the similarity in morphological feature space and latent similarity between target structure and a) morphology (left) or b) joint morphology and structure (right).

same response. There are multiple ways of constraining generated molecules to a particular subspace of chemical space with desirable properties, for instance by including additional reward terms during GFlowNet training. However, one particularly well suited for our approach is, assuming that we have the molecular structure associated with target morphology available (e.g. in the drug analog search setting), conditioning on the joint latent representation. The aim of this is anchoring generated molecules to the known, desirable molecular structure.

We evaluate the capabilities of the proposed approach in constraining searchable space based on the given structure by replacing target latent with a joint representation, generated based on both morphology and associated structure. We evaluate the number of discovered modes (Figure 6), molecular similarity to the given target (Table 1) and number of samples with high predicted assay probability (Figure 7). As can be seen, using joint representations does not impact the ability of GFlowNet to generate high-reward and diverse samples, and actually increases the average number of discovered modes. Crucially, as expected, it also leads to producing more structurally similar to the target molecules, effectively constraining generation process. However, somehow surprisingly, conditioning based on the joint does not seem to increase the proportion of molecules with high assay probability.

Our leading hypothesis is that the joint embedding space learned by the GMC model acquires a stronger structural signal than a morphological signal, thus leading the GFlowNet to sample molecules that are more similar to the target,

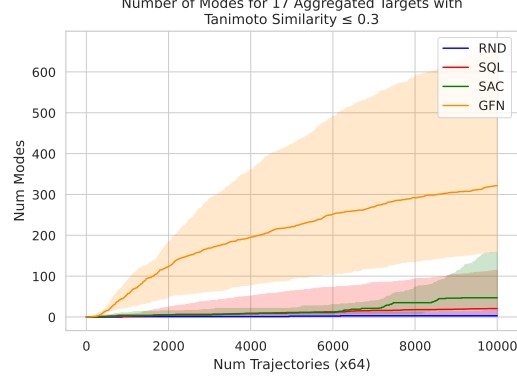

Figure 6. Number of modes discovered for joint morphology and structure-guided generation.

but that don't necessarily trigger the desired morphological profile in the target cell. This claim is supported by Appendix A.6, where the GMC alignment between joint and structural latent space ($r = 0.94$) is greater than the alignment between joint and morphology ($r = 0.88$). Although the alignment between structure and morphology embeddings is even lower ($r = 0.75$), we argue that this setting provides a stronger and more direct signal that guides the GFlowNet towards more diverse molecules that exhibit the desired morphological profile of the target. This is consistent with our results in Figure 7 which suggest that the joint setting yields a higher number of molecules that are structurally similar to the target, but finds less molecules with high assay probabilities than the morphology-only setting.

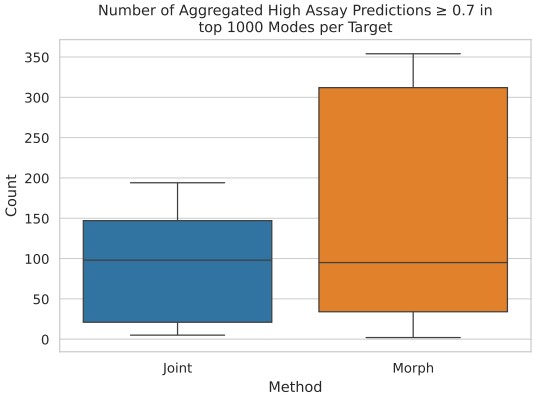

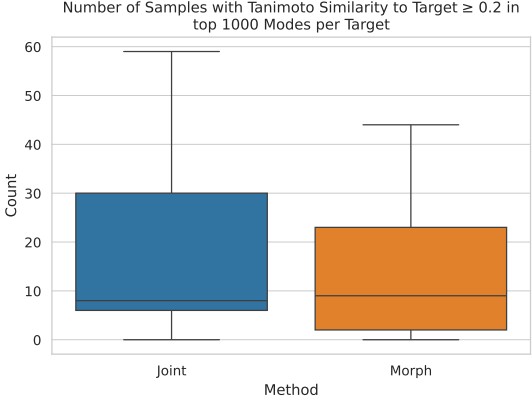

*Figure 7.* Comparison of morphology-only versus joint morphology and structure guided generation: a) number of samples with prediction assay probability $\geq 0.7$ in top 1000 modes (left; higher is better) and b) number of samples in top 1000 modes with Tanimoto similarity to target $\geq 0.2$ (right). Modes are defined as high reward samples with mutual Tanimoto similarity $\leq 0.3$

### A.3. Target selection

To evaluate the generative models, we selected a total of 17 targets from the Cell Painting dataset among which 9 were used for the cluster identity experiments and 8 for the assay activity experiments. All targets were verified to be decomposable into the molecular fragment set used for generative modelling and were chosen to represent a diverse set of morphological clusters and active biological assay.

### A.4. GMC model selection

While training the GMC model, we were faced with the choice of which metric to use for model selection. Other than the standard GMC contrastive loss, we used the correlation between the cross modality distance metric presented in section 4.2 for early-stopping. While there wasn't a significant difference in GFlowNet performance when using these two GMC variants, we found that early-stopping on the correlation metric reduces the variance of cosine similarity between the multimodal GMC embeddings associated

from the same sample when measured in the test split.

### A.5. High Reward Samples

Here we plot some modes sampled by GFlowNet for some targets in a) morphology only guided sampling (left) and b) joint morphology and structure guided sampling (right)

### A.6. GMC cross-modality alignment

Here we plot the correlation between latent distances in one modality versus latent distances in a second modality of the GMC representation space. A high correlation value indicates that the GMC model learns to properly align the modality specific inputs of the data such that associated inputs are aligned closely and distinct inputs aren't. As expected, GMC achieves higher correlation when the joint modality is included in one of the axes since the joint latent space integrates signals from both the structure and morphology features.

## B. Experimental details

In this section, we present the experiment details for the results obtained in the main paper.

### B.1. GMC model training

We follow the specification in the original GMC paper (Poklukar et al., 2022) and select a single model checkpoint for all our experiments by early-stopping on the GMC validation loss. We employ a Graph Convolutional Network (GCN) for the structure encoder and a simple Multilayer Perceptron (MLP) for the cell morphology inputs. MLPs are also used for the projector architecture for all modalities. See table Table 2 for a full breakdown of the hyperparameters we used.

| Parameter | Value |
|---|---|
| Batch size $\beta$ | 128 |
| Number of epochs | 200 |
| Optimizer | Adam |
| Learning rate | $1 \times 2e^{-6}$ |
| Non-Linearity | ReLU |
| Temperature $\tau$ | 0.4 |
| Intermediate Dim. Size $d$ | 1024 |
| Latent Dim. Size $s$ | 1024 |

*Table 2.* Hyperparameters of the Geometric Multimodal Contrastive proxy model

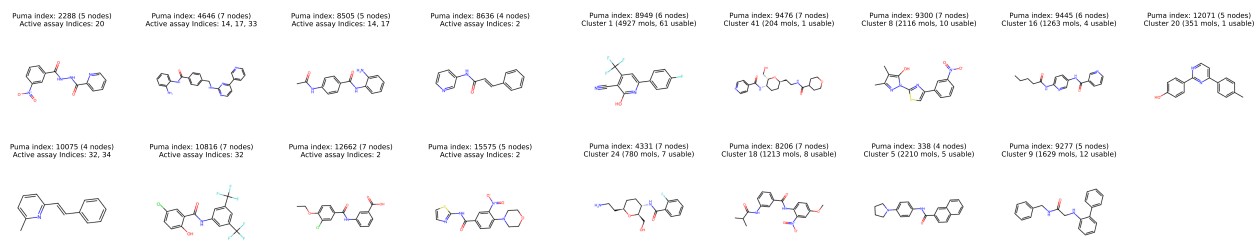

*Figure 8.* Active Assay Targets (left) and Morphological Cluster Targets (right).

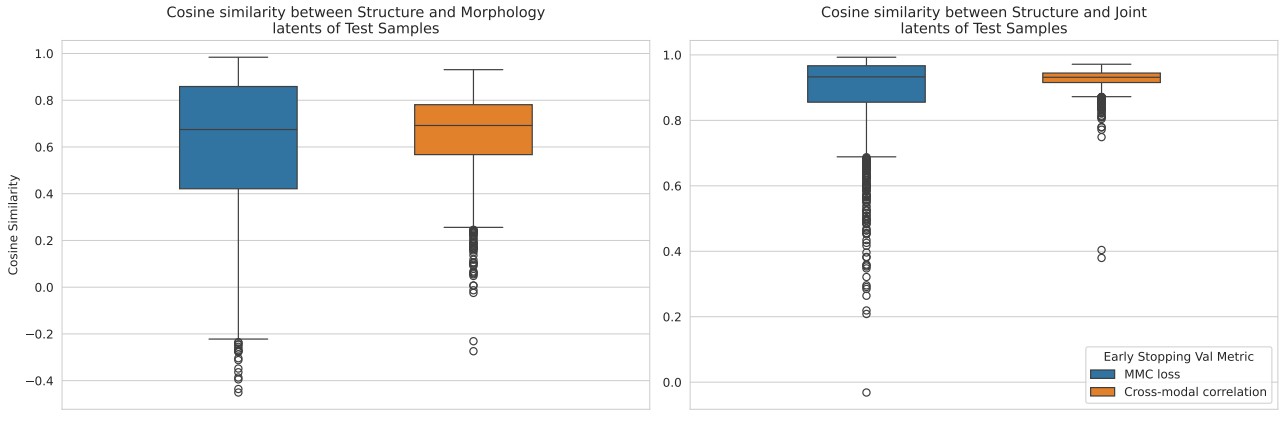

*Figure 9.* Cosine similarity between a) structure and morphology embeddings (left) and b) structure and joint embeddings (right) produced by GMC model on test split.

## B.2. Fragment-based molecule generation

| Parameter | Value |
|---|---|
| Batch size | 64 |
| Number of steps | 10,000 |
| Optimizer | Adam |
| Number of Layers | 4 |
| Hidden Dim. Size | 128 |
| Number of Heads | 2 |
| Positional Embeddings | Rotary |
| Reward scaling $\beta$ in $R^\beta$ | 64 |
| Learning rate | $1 \times 10^{-4}$ |
| $Z$ Learning rate | $1 \times 10^{-3}$ |

*Table 3.* Hyperparameters of the Graph Attention Transformer used across all models in fragment-based molecule generation.

In this section, we provide details on the molecule generation experiments and the hyperparameters we used for the methods presented in the paper. In our experimental setup, we follow the same environment specifications and implementations provided in (Malkin et al., 2022b) with

the exception of a different proxy model (GMC) and reward function. The architecture of the GFlowNet, SAC and SQL models is based on a graph attention transformer (Veličković et al., 2017) whose specification is detailed in table Table 3. For SAC, we use a fixed $\alpha$ value of 0.2 chosen from 0.1, 0.2, 0.5. For SQL, we use a fixed $\alpha$ value of 0.1. All methods use discount factor $\gamma$ value of 1.0.

## B.3. Assay selection

For the selection of assays to train the oracle, we primarily focus on assays that have been linked to morphological features and/or combinations of morphological and chemical properties. We select 37 assays identified in (Moshkov et al., 2023) as predictable from morphological features or combined chemical and morphological features with high accuracy (AUROC > 0.9).

## B.4. Oracle training

We trained two MLPs on molecular fingerprints to predict active biological assays (in a multi-label classification setting), and active morphological clusters (in a multi-class classification setting) for the targets in Appendix A.3. For

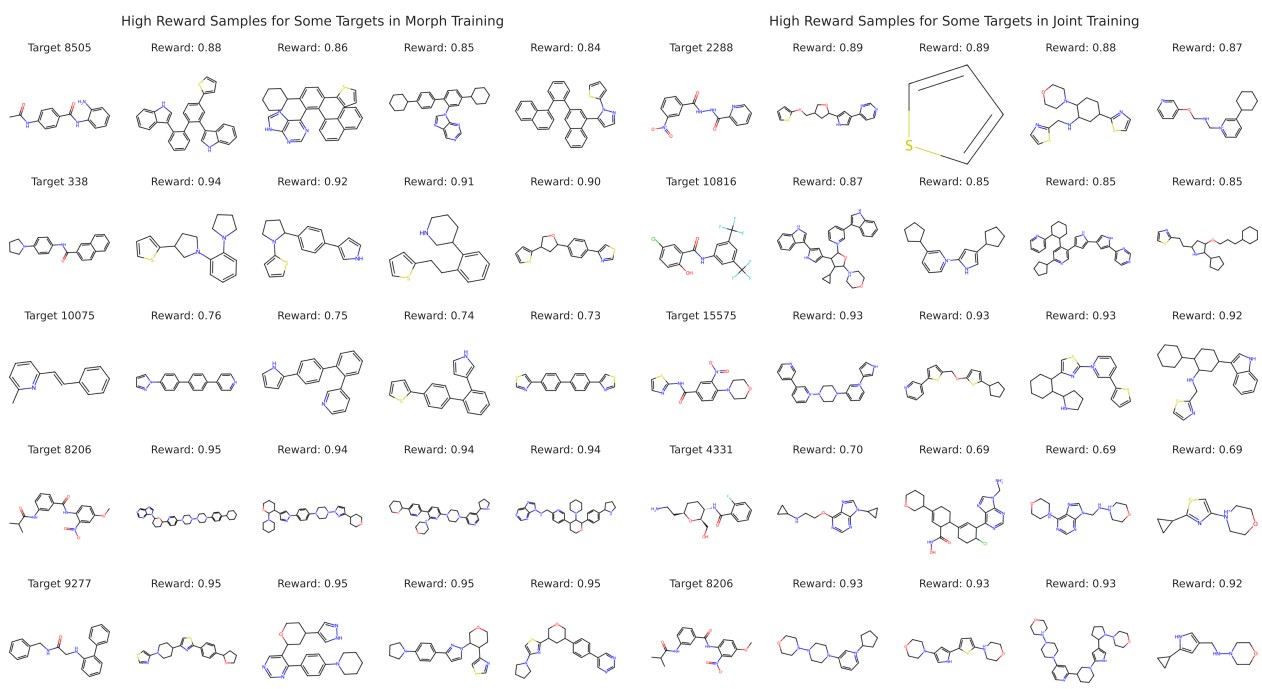

*Figure 10.* High reward samples generated by GFlowNet.

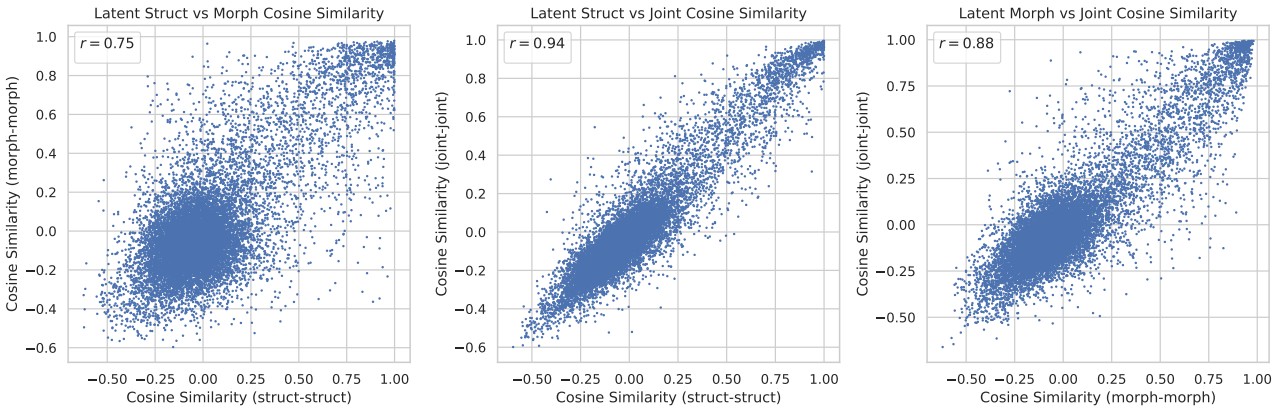

*Figure 11.* Cross-modal alignment of samples.

the input, we use Morgan fingerprints with radius 3 and dimension 2048. The MLP has two 64 dim. hidden layers and uses ReLU activation. Both methods are trained with a learning rate of $1e^{-4}$ with Adam optimizer for 200 epochs. We perform model selection based average precision score on the validation set for both models.