# OpenReview forum: "Cell Morphology-Guided Small Molecule Generation with GFlowNets"
_ICML.cc/2024/Workshop/ML4LMS — ML4LMS Poster_

### Official Review · Reviewer_sZno · 2024-06-01
**Innovative Approach to Drug Discovery with GFlowNets  Brief summary of my review:  This paper presents a novel method for molecular generation using GFlowNets guided by latent representations from a cross-modal contrastive model. It leverages high-content imaging for phenotypic screening to design new molecules with similar biological activities. The approach is significant and innovative, demonstrating high morphological and structural similarity in generated molecules. While the methodology is clear and well-presented, the evaluation on diverse datasets and more details on scalability could enhance the study. Overall, it is a strong contribution to the field of drug discovery.**

**Rating:** 8
**Confidence:** 4

**Review:**

Summary:
The paper presents a novel approach to drug discovery using GFlowNets guided by latent representations from a cross-modal contrastive model. It leverages high-content phenotypic screening and unsupervised multimodal joint embeddings to generate molecules with phenotypic effects similar to target images.

Pros:

Innovative use of GFlowNets for molecular generation.
Effective integration of multimodal data and contrastive learning.
Demonstrates high morphological and structural similarity in generated molecules.

Cons:

Limited evaluation of diverse datasets.
More details on the scalability and computational efficiency would be beneficial.
Evaluation:
The work is original and significant in drug discovery, providing a clear methodology and promising results. The clarity of the presentation is strong, making complex concepts accessible.

---

### Official Review · Reviewer_hCM8 · 2024-06-10
**It has valid points but the connection with biology should be further investigated**

**Rating:** 7
**Confidence:** 4

**Review:**

- the paper presents valid points for the generation and exploration of novel drugs
- the paper presents a good overview of the literature landscape making it clear the objective of the paper
- the paper presents a good application of the GMC loss in drug discovery

Things to improve:

- the main body should contain all the information about the dataset used and the training performed,
- the connection to the biology morphology in my opinion kind of defeats the potential of the paper, it starts by stating "chemical landscape is estimated to be to the 10^60", and then the generation is constrained by high Tanimoto similarity. It would be better to have an assessment of chemical diversity of these generated molecules (have yes high Tanimoto similarity but also different chemical entities - more flat, 3D etc.) you could make a plot showing the different compounds generated and how different they are from one another, as in typical drug discovery pipelines, the lead optimization yes it is based on a template, but the lead identification explores a much vasted chemical space and divesity.

---

### Official Review · Reviewer_6prc · 2024-06-11
**The paper proposed a novel method, using contrastive learning to align molecular graph and mophological image embeddings and using the latent similarity between generated molecules and targeted morphology as reward function to train GFlowNet.**

**Rating:** 8
**Confidence:** 3

**Review:**

Pros:
- Novel approach of using joint contrastive learning latent space similarity as reward to train GFlowNet.
- The outline and rationale are explained clearly.
- The results show high diversity which is promising  to explore the wide molecular space (but also highly variable for different targets, which needs to be experimented further),

Cons:
- The effectiveness of the generated molecules heavily relies on the quality of the joint latent space representations, which is not guarantee with cellpaint assay.
- Lacking negative control (not random sampling): there are many non-active chemical perturbations in public cellpaint datasets that will be important for the model to explore efficiency on.
- The evaluation primarily relies on computational predictions and proxy models for biological activity, which are not sufficient for efficacy and safety of the generated molecules. More extensive biological validation, including experimental assays, would be necessary to confirm the practical utility of this model. However, these are future concerns when the performance could improve significantly.

Overall, the study is (one of) the first attempt at molecular design guided by morphological embedding via GFlowNet. Though the efficiency of the current model is not high, the approach is novel and promising in terms of diversity in generation.